# Bioengineering Approaches to Fight against Orthopedic Biomaterials Related-Infections

**DOI:** 10.3390/ijms231911658

**Published:** 2022-10-01

**Authors:** Joana Barros, Fernando Jorge Monteiro, Maria Pia Ferraz

**Affiliations:** 1i3S—Instituto de Investigação e Inovação em Saúde, Universidade do Porto, R. Alfredo Allen 208, 4200-135 Porto, Portugal; 2INEB—Instituto de Engenharia Biomédica, Universidade do Porto, R. Alfredo Allen 208, 4200-135 Porto, Portugal; 3Departamento de Engenharia Metalúrgica e de Materiais, Faculdade de Engenharia da Universidade do Porto, Rua Roberto Frias, 4200-465 Porto, Portugal

**Keywords:** orthopedic implants, bone infections, bacterial adhesion, bacteria-material interactions, anti-infective biomaterials

## Abstract

One of the most serious complications following the implantation of orthopedic biomaterials is the development of infection. Orthopedic implant-related infections do not only entail clinical problems and patient suffering, but also cause a burden on healthcare care systems. Additionally, the ageing of the world population, in particular in developed countries, has led to an increase in the population above 60 years. This is a significantly vulnerable population segment insofar as biomaterials use is concerned. Implanted materials are highly susceptible to bacterial and fungal colonization and the consequent infection. These microorganisms are often opportunistic, taking advantage of the weakening of the body defenses at the implant surface–tissue interface to attach to tissues or implant surfaces, instigating biofilm formation and subsequent development of infection. The establishment of biofilm leads to tissue destruction, systemic dissemination of the pathogen, and dysfunction of the implant/bone joint, leading to implant failure. Moreover, the contaminated implant can be a reservoir for infection of the surrounding tissue where microorganisms are protected. Therefore, the biofilm increases the pathogenesis of infection since that structure offers protection against host defenses and antimicrobial therapies. Additionally, the rapid emergence of bacterial strains resistant to antibiotics prompted the development of new alternative approaches to prevent and control implant-related infections. Several concepts and approaches have been developed to obtain biomaterials endowed with anti-infective properties. In this review, several anti-infective strategies based on biomaterial engineering are described and discussed in terms of design and fabrication, mechanisms of action, benefits, and drawbacks for preventing and treating orthopaedic biomaterials-related infections.

## 1. Introduction

In orthopedic surgery and traumatology, bone grafting is one of the most frequently performed surgical procedures used for bone-loss repair and bone augmentation [1]. However, prosthetic joints and other orthopedic implant devices (such as pins, screws, plates, and external fixators), required in situations such as osteomyelitis post-debridement, are extremely sensitive to contamination by microorganisms and the subsequent development of infection [2,3].

Orthopedic implant-related infections are among the main reasons for joint arthroplasty and osteosynthesis failure, with severe and devastating outcomes for patients and health systems. Their treatment requires in most cases, the infected implant removal, implant replacement, revision surgeries, or/and amputation, which translates to high rates of morbidity, and increased risk of mortality [4]. In addition to causing significant physical and emotional suffering, implant infections are a massive economic burden to the health systems, estimated to cost more than $8.6 billion annually in the United States and €2 billion in Europe [4,5]. The implant-related infection rate varies according to the type of bone involved (e.g., hip, knee, ankle, or tibia), grade/type of fracture (i.e., closed or open), or type of surgery (i.e., primary or revision) [6,7,8]. For instance, the likelihood of infection following the implantation of a prosthetic hip is 0.3–2.4%, while a total knee replacement is 1–3% [8]. In closed fractures, the incidence of infection after internal fixation is generally low (0.5–2%) when compared to open fractures, wherein the infection rate may exceed 30% [6,7,8]. In revision surgeries concerning implant removal, amputation, or tissue debridement, the risk of infection is higher when compared to primary ones. For instance, the infection rate following total hip arthroplasties is 14.8%, while for total knee revision it is 25.2% [9].

The Centers for Disease Control (CDC) classified biofilms as one of the most pressing clinical obstacles of the century, since they contribute to more than 80% of human bacterial infections [10,11,12]. Biofilms act as communities of microbial cells attached to an inert or living surface that are functionally organized and enclosed in a self-produced polymeric matrix (EPSs), thus contributing to the increasing infection pathogenicity [13].

Over the past 20 years, a wide range of different bioengineering approaches for the treatment of orthopedic implant-related infections has been investigated both in vitro and in vivo with promising results [14,15,16,17,18,19]. However, their implementation in orthopedics is still at an early stage. The present review has as its major goal to revisit the main anti-infective strategies based on biomaterial engineering to overcome orthopedic implant-related infections.

## 2. Implant-Infecting Microorganisms

Implanted materials are highly susceptible to bacterial and fungal colonization and consequent infection [20]. These microorganisms are frequently opportunistic, taking advantage of the weakening of the body defenses at the implant surface–tissue interface, thereby attaching to tissues or implant surfaces, instigating biofilm formation and subsequent development of infection [13,21]. The development of biofilm causes tissue destruction, systemic dissemination of the pathogen and dysfunction of the implant/bone interface, resulting in the failure of implanted material [13,21]. Additionally, the contaminated implant may serve as a reservoir for an infection of the surrounding tissue, where microorganisms may reside intracellularly [13,21].

In Europe and U.S, the most prevalent microorganisms in implant-related infections are Gram-positive bacteria, mainly *Staphylococcus aureus* (33–43%) and *Staphylococcus epidermidis* (17–21%) [13,22], as reflected in Table 1. Other Gram-positive bacteria, such as *Streptococcus viridans* and *Enterococcus* spp. (mainly *Enterococcus faecalis*), are encountered in 1–10% and 3–7% of infections, respectively [13,22,23]. Gram-negative organisms, including *Pseudomonas aeruginosa*, *Escherichia coli*, *Klebsiella pneumonia*, *Proteus mirabilis* and *Proteus vulgaris* are less frequent than Gram-positive, causing around 6% of cases [24]. Anaerobic bacteria (including *Propionibacterium acnes*) and fungi (mainly *Candida albicans*) are also involved on implant-related infections (2–3%) [13,24]. Polymicrobial infections are reported in about 10–11% of the cases; the majority are caused by two bacterial species such as methicillin-resistant *S. aureus* (MRSA) and *Klebsiella* spp. [24,25]. It should be noted that bacteria isolation and identification always depend on the quality of the diagnostic procedure and preceding antimicrobial therapy [24].

In the literature, three types of biomaterials-associated infections are reported: (i) exogenous infection that occurs during or immediately after surgery through direct inoculation into the surgical site; (ii) contiguous infections acquired from spread from an adjacent infectious focus, and (iii) hematogenous infections provided from the distant focus of infection, such as blood or lymph. Concerning the time since surgery and the onset of the infection, these infections can be classified into: (i) early infections (less than three months after the surgery); (ii) delayed infections (between three to 24 months after surgery), and (iii) late infections (more than 24 months after surgery) [29]. Early and delayed infections are commonly caused by trauma or contamination during surgery. *S. aureus*, *Enterococcus* spp., and Gram-negative bacilli, intrinsically virulent microorganisms, are usually the pathogenic agents related to early infections [13,29]. Coagulase-negative *Staphylococci* and *P. acnes*, low-virulent microorganisms, are usually pathogenic agents of delayed infections. Late infections are usually acquired by hematogenous spread, having a considerable range of pathogenic agents originating from the skin, respiratory, dental, or urinary tract infections [13,29,30,31,32].

Methicillin-resistant *S. aureus* (MRSA), Vancomycin-resistant *S. aureus* (VRSA), Methicillin-resistant *S. epidermidis* (MRSE), Vancomycin-resistant *Enterococcus* (VRE) and extended-spectrum β-lactamase-producing *Enterobacteriaceae* (ESBLs) are examples of antibiotic-resistant bacteria that have been commonly linked to implant-related infections, which are a huge threat to human health since they limit therapeutic options to adopt [33,34,35]. In addition, these bacteria are producers of virulence factors (e.g., catalase, hyaluronidase, collagenase, toxins) that play an important role in the degree of severity of the infection, once they promote bacterial adherence to the bone and implant and severe tissue damage [13,21].

## 3. Pathogenesis of Implant-Related Infections

Biofilm formation consists in the irreversible attachment and growth of microorganisms onto surfaces with the concomitant production of the extracellular polymer matrix (EPSs), which alters the microorganism’s phenotypic, growth rate and gene transcription. Biofilms are three-dimensional complex structures that confer significant survival advantages to microorganism communities, which can lead to the recurrence of biofilm-related implant infection. This is a huge concern in the orthopedic field, especially since these communities are highly resilient to host immunity and to conventional anti-microbial therapies [24,36,37,38,39,40,41,42].

Independent-of-the-infection mechanism biofilm formation has four steps (Figure 1): (1) initial adhesion; (2) irreversible adhesion and cell–cell adhesion; (3) proliferation; (4) growth and maturation, and (5) detachment [20,40].

The initial microbial attachment consists of the adhesion of planktonic cells to the implant surfaces or the host through hydrophobic or electrostatic interactions between bacteria and surfaces, mediated by: physical shear forces (e.g., van der Waal, steric interactions, and electrostatic forces); microbial appendages (e.g., pili, flagella, or fimbriae), and adhesion surface proteins (e.g., fibronectin, fibrinogen, vitronectin, thrombospondin, laminin, collagen, von Willebrand factor, and polysaccharides) [20,36,40,43]. Twenty surface-associated adhesins are involved in Staphylococcal biofilm formation. These adhesins mediate initial biofilm attachment and intercellular adhesion during maturation-enhancing cohesion. Autolysins (like At1E) mediated the adhesion of *S. epidermidis* to polymeric surfaces, whereas fibronectin-binding proteins, e.g., FnBPA and FnBPB, induce *S. aureus* invasion into epithelial cells, endothelial cells, and keratinocytes [20,36,40,42,43].

Biofilm maturation consists of microbial proliferation and aggregation, macro- and microcolonies formation, intercellular signaling and quorum sensing (QS). These are mechanisms that induce the expression of specific genes and proteins involved in biofilm structure, virulence and regulation processes [44]. During biofilm maturation, the microbial cells start the secretion of the EPSs and eDNA, that besides stabilizing the biofilm network into the implant surface, are responsible for linkage between clusters, cell-to-cell cohesion, and cellular communication [44]. The main polysaccharide of *S. epidermidis* biofilm matrix is polysaccharide intercellular adhesin (PIA), which aids *staphylococci* in colonizing biomaterial surfaces and protects the proliferating bacteria from polymorphonuclear leukocytes [45]. The microbial microcolonies encased within the EPS communicate between neighboring cells through the QS phenomenon, which controls several physiological processes such as bioluminescence, secretion of virulence factors, biofilm formation and antibiotic resistance [44]. For instance, N-acylated homoserine lactone QS systems of *P. aeruginosa* are responsible for eDNA release and for biofilm structure [46]. The *S. aureus* small peptide named AIP QS activates the *agr*A gene, which regulates the transcription of genes that code for proteases involved in biofilm dispersal [47].

As the biofilm matures, there is an increase in stress-inducing conditions, toxic product accumulation, and limited nutrient availability. These phenomena take the microbial cells to disperse them to other regions of the host’s body or other regions of the medical implant [13,44]. This constitutes the biofilm dispersal phase, where microbial cells (either single cells or clumps of cells) are sloughed off the biofilm. Biofilm dispersal can lead to the dissemination of the detaching microorganisms, which can cause chronic infections, or even reach the bloodstream and cause systemic infections [13,44]. Inhibition of matrix production, enzymatic degradation of EPSs, and surfactant molecules are the mechanisms that contribute to biofilm dispersal. In *staphylococci*, extracellular enzymes (e.g., staphopain cysteine proteases, the V8 glutamyl endopeptidase SspA and staphylococcal nuclease) and Phenol Soluble Modulins (PSMs) have an essential role in the dispersive phase, particularly in implant-associated biofilm infections. This is because these enzymes are able to degrade the biofilm matrix, contributing to the bacterial dispersal and dissemination of biofilm clusters to distal sites [13,44].

Therefore, developing biomaterials that can prevent bacterial adhesion and/or biofilm formation at the implantation site could be an important breakthrough in bone disease treatments.

## 4. Bioengineering Materials as Therapeutic Approaches

The design and fabrication of new antimicrobial approaches based on bioengineering materials remains an important research line in the orthopedic field, given the impact of implanted-related infections on the quality of life of patients and health systems [48].

Various concepts and approaches have been developed to obtain bioengineered materials with anti-infective properties (Figure 2), i.e., capable of preventing microbial adhesion and colonization of bone tissue and implant surfaces, as well as creating a bacteria-free environment around the implant [49,50]. Biomaterials endowed with anti-infective properties need to be tailored according to the specific application. According to their strategy, anti-infective biomaterials can be classified into two main groups: passive surfaces and active surfaces [49,50].

Passive (anti-adhesive) surfaces are those presenting chemistry and/or structure modifications that aim at preventing or reducing bacterial adhesion, without releasing bactericidal agents into the surrounding tissues. Active surfaces have in their composition pre-coated antimicrobial agents such as organic substances (antibiotics and anti-infective peptides, bacteriophages), metals or metal ions (silver, zinc, copper, among others), and/or their combinations, that may inhibit microbial colonization and biofilm formation. The coatings may simply act by delivering high local concentrations of one or more pre-loaded antimicrobials, or they may have a direct or synergistic antimicrobial activity [49,50,51,52,53].

The following sections describe several developed strategies based on biomaterial engineering to fight orthopedic implant-related infections.

### 4.1. Anti-Adhesive Biomaterials

The physicochemical characteristics of the materials integrating the implant surface have a significant impact on microbial adherence. Therefore, material surface chemical composition, surface charge density, surface energy, wettability (hydrophobicity and hydrophilicity), and surface topography may substantially change the implant susceptibility to microbial adhesion and colonization [49,50].

The principle of an anti-adhesive surface development is based on five physical-chemical mechanisms: steric repulsion, electrostatic repulsion, low surface energy, superhydrophobic and hydrophobic interactions, and substrate–microorganism physical interaction [47,53,54,55,56].

Regarding chemical modification, a variety of polymers can be used to create antifouling surfaces, such as poly(ethylene glycol) (PEG), poly-zwitterionic polymers (e.g., poly(2-methacryloyloxyethyl phosphorylcholine), low-energy polymers (e.g., fluoropolymers) and polymers-based hydrogels (e.g., chitosan), as detailed below [54,56,57,58].

PEG is the most employed antifouling polymer for coating implant materials, due to its ability, in an aqueous environment, to form a layer of tightly bound water molecules that act as a physical barrier (steric repulsion) against the attachment of proteins and microorganisms [54,56,57,58]. Francolini et al. produced polyethylene glycol (PEG)-grafted segmented polyurethanes capable of preventing the adhesion of *S. epidermidis* onto their surfaces. Their anti-adhesive property results from the exposure of PEG chains at the material/water interface, enhancing the material bulk and surface hydrophilicity, and consequently the repulsion interactions [58,59].

Due to their higher in vivo stability, poly-zwitterionic materials (highly hydrophilic antifouling polymers) have been used as substitutes for PEG-based antifouling biomaterials [56]. The antifouling properties of poly-zwitterion materials are due to the positive and negative groups incorporated into their structure, as well as the formation of a hydration layer that acts as a physical barrier to avoid protein and bacterial adhesion [56,60,61].

Hydrogels are three-dimensional (3D) porous, highly water-absorbent polymeric networks that can be crosslinked physically or chemically. Hydrogels can be obtained from natural (e.g., chitosan and gelatin) or synthetic polymers (e.g., poly(vinyl alcohol) and poly(sulphobetaine)) [56,58,59,62]. This biomaterial is frequently used in tissue engineering, drug administration, contact lenses, wound dressings, and implantable devices, due to its “soft and wet” characteristics. However, the hydrogels’ poor mechanical qualities generally prevent them from being used in orthopedic implants [56,58,59,62]. Alternatively, coating a thin hydrogel layer onto the device surface is an effective way to combine the advantages of the bulk property of biomedical devices and the biological merit of hydrogels. The attachment and anchoring of hydrogels to implant surface can be performed through several reactions such as surface-initiated radical polymerization, direct photografting, click chemistry, free radical polymerization, photochemical coupling, dopamine-functionalized polymer, self-condensation of silane, and layer-by-layer coating [56,62]. Chitosan (CS) is the most frequently used natural polymer for preparing implant coatings due to its inherent tissue adhesion, antibacterial, and hemostatic properties [58,59,63]. D’Almeida et al. covalently grafted titanium alloy with chitosan through TriEthoxySilylPropylSuccinic Anhydride process to prevent post-surgical infection [63]. That study revealed that CS positively charged quaternary ammonium moieties exerted strong electrostatic interactions on the negatively charged bacterial cell surface, leading to microbial membrane disruption and bacterial death [56,63,64].

Low-surface-energy polymeric coatings (surface energy less than 36 mJ/m^2^ and hydrophobic performance) are considered effective non-sticking surfaces due to their chemical inertness and non-wetting properties [65,66,67]. For instance, Song et al. have shown that the fluorine components, introduced into poly(butyl methacrylate-co-ethylene dimethacrylate) films, caused low surface energy and hydrophobicity, thus preventing *E. coli* and *S. aureus* adhesion [67].

Several studies have focused on obtaining material surfaces with extreme wettability, super-hydrophobic (Ɵ > 150°) or super-hydrophilic (Ɵ < 5°), as potential anti-infective surfaces [68,69,70,71,72]. Super-hydrophobic surfaces can present bacterial repellency properties through their high surface roughness and low surface energy that create a stable or metastable air layer at the material surface (called the “lotus effect”), where debris and pathogens are removed as water contacts and subsequently rolls off the surface [68,69,70,71,72]. For instance, Naderizadeh et al. showed that the adhesion of *E. coli*, *S. aureus,* and *P. aeruginosa* was significantly reduced on the superhydrophobic coatings based on biomass-derived bioresin polyfurfuryl alcohol (PFA) [71]. Stallard et al. reported that superhydrophobic surfaces based on siloxane and fluorinated siloxane elastomeric coatings exhibit antimicrobial properties and significantly reduce bovine serum albumin and bovine fibrinogen adsorption [68]. On the other hand, extremely water-attractive surfaces can also exhibit anti-infective performance via their super-hydrophilicity, which is able to form a dense layer of water molecules, weakening the interaction between bacteria and substratum [68,69,70,71,72]. Choi et al. produced super-hydrophilic interfaces by layer-by-layer (LbL) assembly of the biotic materials chitosan (CHI) and rice husk ash (RHA) nanosilica with anti-adhesive properties, reducing the attachment of proteins, as well as of *S. aureus* and *P. aeruginosa* [73].

Another method to reduce bacterial adhesion to biomaterials consists of coating of the surfaces with serum, plasma, or protein solutions [74,75]. These solutions cause changes in the physicochemical properties of the biomaterial’s surfaces, interfering with bacterial adherence and host adhesins adsorption [49,66,67,74,75,76,77,78,79]. An et al., using a rabbit model, showed that albumin-coated implants presented a lower infection rate than non-coated implants [64,74,80,81,82]. This inhibition can be explained through binding to the bacterial receptors or by changing the substratum surface to a more hydrophilic behavior [74,75].

The ability of a microbial cell to remain attached to a surface reflects the nature of non-covalent interactions between the substrate and cell wall functional groups during the initial attachment phases [81]. According to attachment point theory, the biomaterial structure and morphology can be modified to produce anti-adhesive surfaces. Several patterning techniques (e.g., lithography, reactive ion etching, femtosecond laser writing, electro-chemical oxidation, electron beam evaporation, micro-contact printing, hot embossing and microfluidics) can be employed to fabricate antifouling structured surfaces [81]. These strategies can be applied whether at the micro- and/or nano-metric levels or nanometer-size combined with the micro-patterns. The antifouling phenomenon from physical modifications of material surfaces could be due to the: (i) surface roughness size (below 0.2 μm) that induces low susceptibility to bacterial attachment; (ii) micrometric and nanometric pillars, peaks and valleys of the implant surface that affect the organization of bacterial cells and their intracellular transduction signaling pathways, and (iii) changing of surface properties such as wettability [80,81,83,84]. It should be noted that the nanostructures are important parameters in the production of antifouling surfaces since effective air entrapment in the three-dimensional nanomorphology renders these surfaces superhydrophobic and slippery. For instance, Pucket et al. showed that on nano-rough titanium surfaces produced by electron beam evaporation, the amount of adhered *S. aureus*, *S. epidermidis* and *P. aeruginosa* was lower when compared to conventional titanium surfaces [83]. Moreover, they observed that the hydrophilicity of nano-tubular and nano-textured titanium surfaces, produced through anodizing processes, also contributed to increasing the antifouling surfaces [83].

### 4.2. Active Biomaterials

Active biomaterials are based on pre-coated surfaces with antimicrobial agents that can be: (i) organic compounds (e.g., antibiotics, antimicrobial peptides, QS inhibitors or bacteriophages), and (ii) inorganic compounds (mainly metal ions, e.g., silver, gold, zinc, copper, magnesium), as detailed in the sections below. These surfaces are interesting in bone implant-related infections due to their capability to kill pathogens upon contact. According to their functional principle, surfaces can be categorized as either active-releasing surfaces or contact-active surfaces. Active-releasing surfaces result from the entrapment or coating of an antimicrobial agent in the bulk or in the coating of a biomaterial, which will be released upon interaction with its surrounding environment and/or stimuli, killing the planktonic and sessile microorganisms. Conversely, contact-active surfaces result from the covalent immobilization of an active agent on the implant’s surface, killing microorganisms that attempt to adhere [49,50,51,52,53]. Some research data illustrating the efficiency of active biomaterials in battling infections associated with bone-related biomaterials are included in Table 2.

In general, the antimicrobial substances (organic or inorganic) can be incorporated by mixing them with materials during production, absorbing them after production in the case of permeable or porous biomaterials, covalently binding them to functionalized coatings, and incorporating them into self-assembled mono/multilayer organic coatings [49,50,51,52,53]. Drug release occurs mainly by the means of the following: diffusion to the aqueous phase; erosion/degradation of resorbable loaded matrices, and hydrolysis of covalent bonds. The active principle release kinetics depends on the molecular bonds and the biodegradation/bioerosion rate, therefore it is possible to obtain several delivery kinetics profiles [49,50,51,52,53].

#### 4.2.1. Biomaterials with Anti-Infective Organic Agents

Antibiotics (e.g., vancomycin, daptomycin, rifampicin, amoxicillin, levofloxacin, gentamicin or linezolid) are widely employed for the prevention and treatment of peri-prosthetic infections [123]. Up to now, despite wide research on several antibacterial surfaces, antibiotic-loaded implant materials are the only approach that has reached the market [85,87,89,90]. However, this approach has some drawbacks: (a) dose-dependent antibiotic activity; (b) delivery of sub-therapeutic antibiotic levels, favoring bacterial resistance development (c) limited antibiotic diffusion into peri-implant tissues, and (d) systemic and local cytotoxicity, impairing bone growth and implant osseointegration [86,88,124].

Therefore, it is essential to have alternative antimicrobial solutions to solve the aforementioned issues as well as to effectively prevent and manage implant-related infections.

Antimicrobial peptides (AMPs) are an interesting group of anti-infective agents currently viewed as alternatives to mitigate the problem of antibiotic-resistant microorganisms. Such peptides combine the antimicrobial activity against a wide range of pathogens (Gram-positive and Gram-negative bacteria, fungi, parasites and enveloped viruses) with acceptable biocompatibility [125,126,127,128]. AMPs are positively charged and present amphipathic behavior which enables interactions with bacterial membranes (lipopolysaccharides in Gram-negative and teichoic acids in Gram-positive bacteria), disrupting bacterial membrane integrity and leading to bacterial lysis [77,79,94,95,129,130,131]. It should be noted that AMPs hardly lead to resistance development [128]. Therefore, AMPs can represent excellent coating agents with broad-spectrum antimicrobial activity for preventing implant-associated infections [132,133,134]. AMPs coating onto implant surfaces is based on three strategies: (i) direct coating of peptide sequences onto biomaterials [93]; (ii) covalent linkage of peptides onto biomaterials using functional groups [91] and (iii) integration of peptides into a matrix layer or scaffold to be released over time [92]. In the latter method, peptides are initially encapsulated in a hydrogel or matrix, after which this hydrogel or matrix is applied to the biomaterial’s surface as a layer [128]. Independently of the chosen method, AMPs coating density on biomaterials is always a large challenge. In addition, crucial factors such as the peptides’ length, flexibility, and orientation, as well as spacer molecules that bind the peptides to the surface, must be taken into account for a coating to be successful and effective [128]. Besides, although several biophysical techniques exist that prove the existence of an AMP layer on the material, it is still difficult to pinpoint exactly how many molecules are attached. Moreover, AMPs are expensive to produce and are vulnerable to pH fluctuations and proteases in the environment [77,95,128].

As previously mentioned, the bacterial behavior within biofilms, bacterial biofilm resistance to external conditions and the virulence pathway of bacteria are regulated by the QS phenomenon [104,135]. Several studies have shown that QS inhibitors can effectively reduce the growth of planktonic bacteria and effectively inhibit/disintegrate bacterial biofilms, so as to achieve the effect of treating bacterial infection [102,103,104,135,136]. Therefore, these inhibitors interfere with bacterial communication, blocking the QS-mediated pathogenic infection by shutting down the expression of the pathogenic gene and diminishing the expression of virulence factors [96,98,137,138]. *Staphylococcal* infections are the primary cause of orthopedic implant failure, and the accessory gene regulator (*agr*) QS system is a crucial regulator of their pathogenic phenotype. For instance, Agr antagonist TrAIP-II or the AI agonist AIP-I interferes with QS communication or biofilm dispersal, respectively [99,101,135]. Several anti-Agr compounds have been studied as alternative agents for anti-infective biomaterials formulation [100,135]. Several studies have shown that dihydropyrrolones (DHPs), derivatives of the fimbrolide class of QS inhibitors, covalently immobilized on biomaterial surfaces, can diminish a microorganism’s colonization and avoid biofilm formation [99,101,135]. Nevertheless, there are still some drawbacks related to this approach, such as a narrow target spectrum; bacteria employing unique QS systems; the lack of large-scale clinical QS inhibitors testing, and the possibility to develop bacterial resistance toward QS inhibitors [135].

Bacteriophage (phage) therapy has emerged as a potential alternative therapy to conventional antibiotics to manage and treat biofilm-related infections [139,140,141,142,143,144]. Lytic phages, natural antimicrobial agents, are viruses harmless to humans that infect and kill target bacteria. They are specific to a bacterial host, replicating exponentially inside the bacterial cell. At the end of their life cycle, newly formed phages are released to infect new bacterial targets [142,145]. Additionally, phages encode in their genome some important hydrolytic enzymes (hydrolases and lyases) that degrade extracellular polymeric substances (EPSs) present in biofilms, making them a powerful weapon against biofilm-related infections [140,142]. Several pre-clinical animal studies support the use of phage therapy in the clinical treatment of infections, such as implant-related, diabetic cutaneous wounds, peri-prosthetic joint osteomyelitis and soft tissue infections [144,146,147]. These studies showed that the administration of phage solutions to the site of infections resulted in biofilm burden reduction and improvement of treatment outcomes [144,146,147]. Biomaterial-based local phage delivery systems have emerged as a therapeutic option in managing bone and joint infections because, besides conferring phage protection and stability, they allow for attaining phage delivery at or close to the infection site, phage bioavailability and prolonged residence at the infected site [17,34,105,106,107,148]. In general, three strategies can be used to produce phage-loaded biomaterials: (1) embedding; (2) encapsulation, and (3) surface functionalization through phage adsorption or covalent binding. The strategy choice is dictated by biomaterial type and its processability [149]. For instance, ceramics such as hydroxyapatite (HAP) or other calcium phosphates (CAP) require high sintering temperatures during processing, which dictates that phage adsorption after sintering is the only viable method [97]. One of the most commonly used strategies is the encapsulation of phages into hydrogels, liposomes or fibers. Barros et al. encapsulated phages into the alginate-nanohydroxyapatite hydrogel, showing that this delivery system can be a successful strategy in the management of bone implant-related infections by providing phage release within a broad range of pH (5 to 9). This strategy leads to promising features for phage delivery in the bone environment, with adequate tissue response and biosafety profiles, strong osteogenic and mineralization response, and excellent antimicrobial activity by inhibiting the attachment and colonization of multidrug-resistant bacteria surrounding and inside the femoral tissues [17]. In 2019, the Food and Drug Administration (FDA) approved the first United States (US) clinical trial for intravenous phage therapy [108]; however, there are still limitations, such as the production of robust and standardized phage preparations, the limited number of in vivo pharmacokinetic and pharmacodynamic studies, and so more testing is required to allow phage therapy to become a standardized and well-accepted strategy in clinical practice.

Figure 3 represents the main biomaterials strategies based on coating organic agents to avoid contamination and subsequent orthopedic implant-related infection.

#### 4.2.2. Biomaterials with Anti-Infective Inorganic Agents

Silver, gold, zinc oxide, titanium dioxide, magnesium oxide, or copper oxide are inorganic metallic nanoparticles (NPs) that have been deemed to be effective antimicrobial agents to fight against biofilm-related infections [119]. The antimicrobial effectiveness of NPs are typically correlated with their capacity to: penetrate and disrupt the membrane of the microbial cells via membrane-damaging abrasiveness; reduce the cell’s permeability, and induce antimicrobial effects within the cells (e.g., producing reactive oxygen species (ROS), nucleic acids and protein interactions, enzyme inactivation and efflux pump overexpression) [119,150]. Besides, NPs can also cross the reticuloendothelial barrier, improving the antimicrobial agent’s internalization and the distribution of both hydrophilic and lipophilic molecules [119,150]. The physicochemical characteristics of NPs, including their chemistry, particle size and shape, surface charge and zeta potential, solubility, stability, and surface-to-volume ratio, have a direct impact on the antimicrobial capability of these materials [117,119,151]. For instance, NPs’ surface charge can affect cellular biodistribution and uptake once the NPs’ charges regulate interactions with tissues and tissue constituents. Although hydrophilic NPs have longer blood circulation due to reduced interactions with opsonins, hydrophobic NPs can modulate the interactions with the phospholipid layer within the microbial membrane [119]. There are three major ways to incorporate inorganic metallic nanoparticles into biomaterials: (1) integration into biomaterial matrix; (2) production of a film coated with inorganic NPs, and (3) grafting/immobilization onto the biomaterial surface [119]. As described by Spirescu et al., the sorts of matrices employed for NPs-loaded biomaterials development include a variety of materials, from natural or manufactured polymers to carbonaceous materials, such as graphene [119].

Silver (Ag) NPs are used for coatings of biomaterials due to their antimicrobial effectiveness against a wide assemblage of microorganisms, such as fungi, viruses, Gram-negative and Gram-positive bacteria including drug-resistant bacteria [110,111,152,153,154,155]. Although the exact mechanism of action of Ag^+^ ions is still not known, there are various hypothesized mechanisms, including direct damage to microorganism membrane, ROS formation, and disruption of ATP synthesis and DNA replication [109,156,157]. Additionally, to Ag^+^ ions, the morphology, size and shape of nanoparticles themselves, which determine their physicochemical properties and Ag^+^ release kinetics, also contribute to Ag NPs antimicrobial efficiency [158,159]. For instance, NPs with smaller diameters and greater surface areas contribute to increased antimicrobial bioactivity because they lead to the production of more ROS, necrotic factors, and apoptotic agents [119]. Ag NPs generally bind to the microbial cell membrane, and inside microorganisms interact with phosphorous- and sulfur-containing compounds (e.g., DNA), causing microbial death. The respiratory chain and cell division are first affected by the Ag NPs [110]. However, high Ag^+^ ions concentrations may have harmful effects on human cells. Therefore, the use of Ag NPs should be carefully regulated [119,160].

Gold (Au) NPs are stable metallic particles with intrinsic antibacterial and antifungal activity [109,112,113,114,119,160]. It should be noted that Au NPs exhibit in general higher antibacterial activity against Gram-negative bacteria probably due to their thinner cell walls and more stable electrostatic contacts [119,160]. The antimicrobial effects of Au NPs mechanisms include: (i) interactions between NPs and the microbial cell wall guided by electrostatic forces and carbohydrate, lipid, and protein binding; (ii) damages in the microbial cell membrane and wall with subsequent ribosome and mitochondrion impairment; (iii) inhibition of thiol groups present within microbial cells; (iv) intracellular ROS concentration increase and (v) microbial cell lysis [119,160,161,162]. The size and functionalization of Au NPs have a direct impact on their antimicrobial activity; as a result, NPs with smaller average sizes have enhanced antimicrobial performance. Moreover, the Physicochemical characteristics of the capping materials also influence their activity, once they regulate NPs’ surface characteristics [119,163]. However, cytotoxicity and biocompatibility problems have been linked to the use of Au NPs, making it necessary to strike a balance between toxic effects and antimicrobial activity when designing Au NPs [119,163].

The biocompatibility of zinc oxide (ZnO) NPs makes them attractive for many biomedical applications, such as tissue engineering, drug delivery systems, antimicrobial coatings, bioimaging, and antioxidant agents [116,117,118,119]. ZnO NPs have been explored as antimicrobial agents against bacteria, fungi and viruses by several authors due to their attractive properties such as large surface area, reduced size, high surface reactivity, and ability to absorb UV radiation [116,117,118,119]. In general, ZnO NPs’ antimicrobial action mechanisms include: (i) cell membrane structure deformation and consequent loss in cell integrity; (ii) disruption of metabolic and enzymatic processes and/or (iii) ROS production [119,153,164,165]. However, the release of Zn^2+^ ions depends on the NPs physicochemical and morphological properties. Furthermore, antimicrobial features are also influenced by the microbial strain, NPs concentration, and time of interaction [119].

Titanium dioxide (TiO_2_) NPs, due to their photocatalytic antimicrobial activity, are the most studied NPs as an alternative antimicrobial agent [121]. TiO_2_ NPs generate ROS, such as hydrogen peroxide and hydroxyl radicals, that upon exposure to ultraviolet (UV) radiation, lead to damage to microbial cell membranes, compromising membrane semipermeability, interfering with oxidative phosphorylation, and causing microbial cell death [120,153]. Similarly to ZnO NPs, TiO_2_ NPs main characteristics that affect their antibacterial activities are the shape, size, crystal structure, surface charge, chemistry, concentration, and exposure time [119].

Like to TiO NPs, MgO NPs have been approved by the FDA as safe materials, which has sparked curiosity among scientists about their potential use in biomedical fields. MgO NPs present antibacterial activity against Gram-positive and Gram-negative bacteria (e.g., *S. aureus*, *P. aeruginosa*, and *E. coli*) and fungi (e.g., *Candida albicans*), as well as anti-biofilm characteristics [122]. The antimicrobial mechanisms underlying MgO NPs are based on (1) the dissociation of Mg^2+^ ions, which generate the superoxide anion through the reaction with oxygen present on the microbial cell surface, and (2) ROS production, which induces disruption of the membrane and causes cell death. MgO NPs physicochemical properties such as surface area, chemistry, roughness, and wettability, could also interfere with the bacterial QS, thus inhibiting biofilm formation [119].

Copper oxide (CuO) NPs have been used as competitor antimicrobial agents to Au NPs and Ag NPs, due to their low production costs involved. In general, the antimicrobial bioactivity of Cu NP results from the formation of the Cu+ ions complexes and the production of ROS, processes that lead to the inactivation of microbial enzymes and disturbances of amino acid and nucleic acid biosynthesis, respectively. However, some cytotoxicity and genotoxic effects of Cu NPs have been reported, namely when used at high concentrations, thereby more studies should be performed to ensure their safety [119,166].

## 5. Conclusions and Final Remarks

Any surgical intervention, particularly ones including the implantation of biomaterials, has a high risk of associated infections. These infections can be devastating for the patients and overload healthcare systems.

Currently, prophylactic systemic antibiotic therapy is administered as a preventive and therapeutic measure to patients to whom an implant is applied. However, this therapy entails many disadvantages, including the low drug concentration reaching the target site and the corresponding limited antimicrobial activity at the target site. Moreover, the wide variety of pathogenic bacteria causing infections related to orthopedic implants, the emergence of bacteria resistant to antibiotics and the cytotoxicity associated with them lead to the need to search for new alternative approaches to prevent and control implant-related infections. The use of local strategies such as anti-adhesive surfaces or active surfaces has emerged as an effective approach to preventing and treating orthopedic biomaterial-related infections. Various concepts and approaches have been applied in the development of bioengineered materials with anti-infective properties, preventing microbial adhesion and colonization into bone tissue, and implant surfaces, and/or creating a free-bacteria environment around the implant. Biomaterials endowed with anti-infective properties need to be tailored according to the specific application and can be classified into two main groups: passive surfaces presenting chemistry and/or structure modifications which prevent or reduce microbial adhesion; and active surfaces with pre-entrapped or coated antimicrobial agents (antibiotics, peptides, bacteriophages, metals or metal ions), which will be released upon interaction with its surrounding environment and/or stimuli, killing the planktonic and sessile microorganisms.

A variety of polymers (e.g., PEG or polymer-based hydrogels), solutions (serum, plasma, or protein) and physical modifications (structure or morphology) can be used to create passive surfaces. Their anti-fouling mechanism is based on physicochemical mechanisms (e.g., steric repulsion, electrostatic repulsion, low surface energy, superhydrophobic and hydrophobic interactions, and substrate-microorganism physical interaction), which act as a physical barrier to avoid protein and bacterial adhesion.

Active biomaterials are based on the incorporation of organic (e.g., antibiotics, antimicrobial peptides, QS inhibitors, or bacteriophages) and inorganic (mainly metal ions, e.g., silver, gold, zinc, copper, magnesium) antimicrobial agents into the biomaterial surface. The antimicrobial agents can be covalently bound to functionalized coatings, or incorporated into self-assembled mono/multilayer coatings during or after biomaterial production.

Antibiotics (e.g., vancomycin, daptomycin, rifampicin, amoxicillin, levofloxacin, gentamicin, or linezolid) are widely employed for the prevention and treatment of peri-prosthetic infections. Up to now, antibiotic-loaded implant materials are the only approach that has reached the market. However, this approach has some drawbacks namely: dose-dependent activity; sub-therapeutic delivery, limited diffusion into peri-implant tissues, and systemic and local cytotoxicity.

Antimicrobial peptides (AMPs) are an interesting group of anti-infective agents currently viewed as alternatives to mitigate the problem of antibiotic-resistant microorganisms. However, AMPs are expensive to produce and are vulnerable to pH fluctuations and proteases in the environment.

QS inhibitors show interesting antimicrobial properties, namely in the inhibition and disintegration of biofilms. Nevertheless, some drawbacks are reported, such as a narrow target spectrum, specific QS for target bacteria, the lack of large-scale clinical QS inhibitors testing, and QS-resistant bacteria development.

Phage therapy has emerged as a potential alternative therapy to conventional antibiotics to manage and treat biofilm-related infections. However, more tests such as standardized and robust phage production, in vivo pharmacokinetic and pharmacodynamics studies, are required in order to allow phage therapy to become a standardized and well-accepted strategy in clinical practice.

Silver, gold, zinc oxide, titanium dioxide, magnesium oxide, or copper oxide are in-organic metallic nanoparticles (NPs), that have been deemed to be effective antimicrobial agents to fight against biofilm-related infections; however, there are still several side effects to be controlled concerning its use in clinical applications.

In sum, a variety of concepts and approaches are applied to create anti-infective bioengineered materials to combat infections associated with orthopedic biomaterials. However, there is still a long way to go before they are used in clinical settings, requiring a search for new approaches or the improve of the available ones in order to overcome safety and efficacy issues related to their implementation.

## Figures and Tables

**Figure 1 ijms-23-11658-f001:**
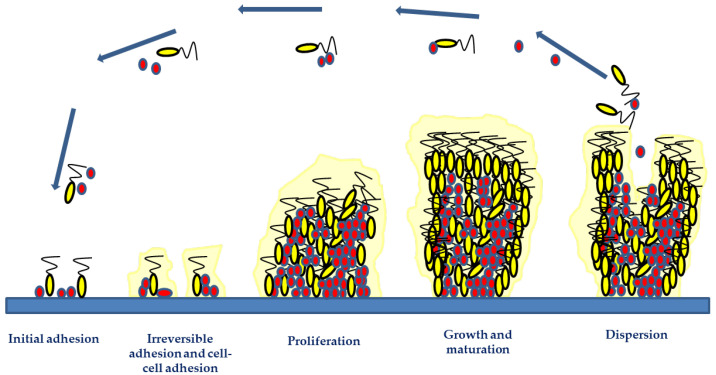
Biofilm development phases include initial adhesion, irreversible adhesion and cell–cell adhesion, proliferation, growth and maturation, and detachment. The two initial phases involve the attachment of microorganisms via hydrophobic or electrostatic interactions to implant surfaces and their involvement in cell-to-cell bindings. The microorganism growth and accumulation during the proliferation and maturation phases result in the development of a mature biofilm structure. Adhesive and disruptive processes occur during the biofilm maturation phase. The final stage of biofilm formation is the detachment phase, involving microbial dispersal and dissemination, which may lead to new infection foci.

**Figure 2 ijms-23-11658-f002:**
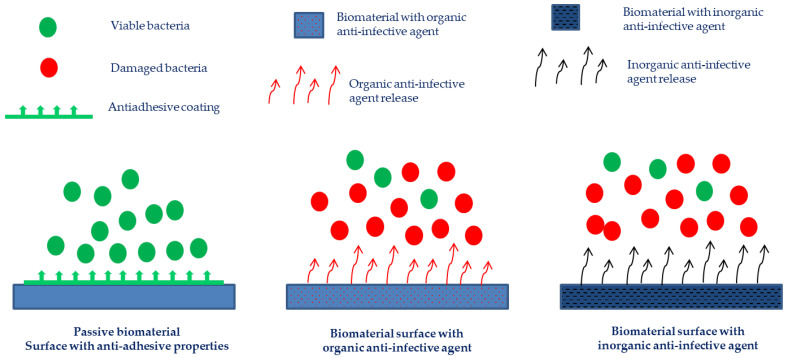
Different strategies have been developed to trigger anti-infective activity in biomaterials.

**Figure 3 ijms-23-11658-f003:**
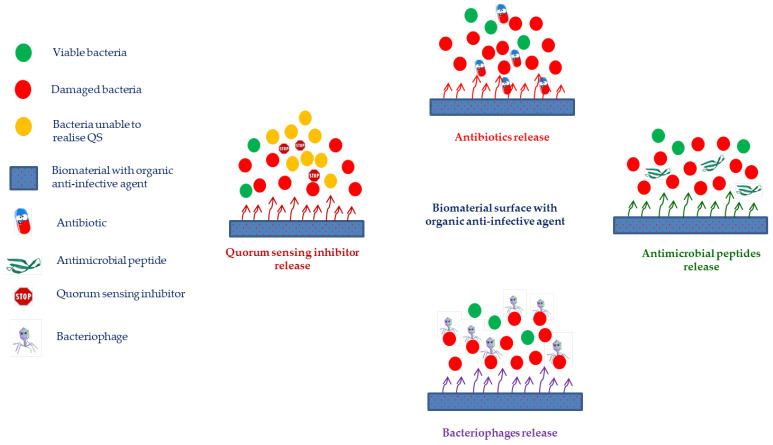
Several strategies developed to trigger antibacterial action in biomaterials involving anti-infective organic agents.

**Table 1 ijms-23-11658-t001:** Prevalence of implant-infecting bacteria in Europe and the U.S. according to the implant type and site.

Species	Prevalence in Knee Arthroplasty Infections (%)	Prevalence in Hip Arthroplasty Infections (%)	Prevalence in Infections Involving External Fixation (%)	Prevalence in Infections Involving Internal Fixation (%)	References
*S. aureus*	26.4	24.4	47.8	42.5	[13,26,27,28]
*S. epidermidis*	41.8	43.6	15.2	21.9	[13,26,27]
*E. faecalis*	2.6	3.5	8.7	5.3	[13,27]
*P. aeruginosa*	4.4	3.7	14.1	4.3	[13,27]
*E. coli*	5.3	n/d	n/d	n/d	[13,26,27]

n/d—not defined.

**Table 2 ijms-23-11658-t002:** Some examples of active biomaterials in fighting bone-related biomaterial infections.

	Anti-Infective Agents	Type of Study	References
Biomaterials with organic agents	Antibiotics	*In vitro*	[85,86,87,88]
		*In vivo* (rabbits)	[89]
		Clinical trial	[90]
	Antimicrobial peptides	*In vitro*	[77,79,91,92,93]
		*In vivo**/*Clinical trial	[94,95]
	Quorum-sensing inhibitors	*In vitro*	[17,96,97,98,99,100,101,102,103,104]
	Phages	*In vitro* Clinical trial	[17,34,105,106,107,108]
Biomaterials with inorganic agents	Silver nanoparticles	*In vitro*	[109]
		Clinical trial	[110,111]
	Gold nanoparticles	*In vitro*	[109,112,113,114,115]
	Zinc oxide nanoparticles	*In vitro*	[116,117,118,119]
	Titanium dioxide nanoparticles	*In vitro*	[120]
		*In vivo* (mice)/Clinical trial	[121]
	Magnesium/Copper oxide nanoparticles	*In vitro*	[122]

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
