# Peer review of "Bioengineering Approaches to Fight against Orthopedic Biomaterials Related-Infections"

_ijms, 2022, doi:10.3390/ijms231911658_

Round 1
Reviewer 1 Report
The manuscript “Bioengineering approaches to fight against orthopedic biomaterials related-infections ” is a review describing the most common types of pathogens responsible for post surgical infections, as well as the involved mechanisms and various methods of improving the antibacterial properties of ortopedic implants.
A great number of research articles and review articles are constantly being published on the topic of improving the antibacterial properties of ortopedic implants, hence the topic is of current interest.
A review as such is intended for a broader audience, mainly new researchers in the field or non-experts and this review uses a clear and comprehensive language.
In Section 1 there are presented some general statistics of the recurrence of post-implant bacterial infections, building on the premise of the article.
In Section 2 there are presented the most common types of bacteria found to be responsible for implant site infection.
In Section 3 are presented the mechanisms involved in bacterial attachment, development and spreading.
In Section 4 are presented various antibacterial materials and strategies that have been tested on implants.
All the presented information is scientifically sound. The figures are easy to understand and they reflect the written information.
The conclusions are accurate, although here is a weak point in this work, since they are based mainly on antibiotic activity, however partial conclusions are implied at each subsection in Section 4.
Another weaker spot of this work is the inclusion of references that are a bit old.
I have read this review with pleasure and I do not have any comments or remarks, so I recommend your manuscript for publication in present form.
Author Response
Reviewer 1
Comment: The manuscript “Bioengineering approaches to fight against orthopedic biomaterials related-infections” is a review describing the most common types of pathogens responsible for post-surgical infections, as well as the involved mechanisms and various methods of improving the antibacterial properties of ortopedic implants.
A great number of research articles and review articles are constantly being published on the topic of improving the antibacterial properties of ortopedic implants, hence the topic is of current interest.
A review as such is intended for a broader audience, mainly new researchers in the field or non-experts and this review uses a clear and comprehensive language.
In Section 1 there are presented some general statistics of the recurrence of post-implant bacterial infections, building on the premise of the article.
In Section 2 there are presented the most common types of bacteria found to be responsible for implant site infection.
In Section 3 are presented the mechanisms involved in bacterial attachment, development and spreading.
In Section 4 are presented various antibacterial materials and strategies that have been tested on implants.
All the presented information is scientifically sound. The figures are easy to understand and they reflect the written information.
The conclusions are accurate, although here is a weak point in this work, since they are based mainly on antibiotic activity, however partial conclusions are implied at each subsection in Section 4.
Another weaker spot of this work is the inclusion of references that are a bit old.
I have read this review with pleasure and I do not have any comments or remarks, so I recommend your manuscript for publication in present form.
Answers for reviewer 1:
Dear reviewer
Thank you very much for your helpful and important comments. They were taken into consideration and responded to below and marked on the manuscript in yellow. Besides, some language improvements were made to the manuscript and were highlighted in yellow.
The conclusions were modified according to your comments (Pages 13-14, section 5, lines 581-630, marked in yellow).
“Various concepts and approaches have been applied in the development of bioengineered materials with anti-infective properties, preventing microbial adhesion and colonization into bone tissue, and implant surfaces, and/or creating a free-bacteria environment around the implant. Biomaterials endowed with anti-infective properties need to be tailored according to the specific application and can be classified into two main groups: passive surfaces presenting chemistry and/or structure modifications which prevent or reduce microbial adhesion; and active surfaces with pre-entrapped or coated antimicrobial agents (antibiotics, peptides, bacteriophages, metals or metal ions), which will be released upon interaction with its surrounding environment and/or stimuli, killing the planktonic and sessile microorganisms.
A variety of polymers (e.g. PEG or polymer-based hydrogels), solutions (serum, plasma, or protein) and physical modifications (structure or morphology) can be used to create passive surfaces. Their anti-fouling mechanism is based on physicochemical mechanisms (e.g. steric repulsion, electrostatic repulsion, low surface energy, superhydrophobic and hydrophobic interactions, and substrate-microorganism physical interaction), which act as a physical barrier to avoid protein and bacterial adhesion.
Active biomaterials are based on the incorporation of organic (e.g. antibiotics, antimicrobial peptides, QS inhibitors, or bacteriophages) and inorganic (mainly metal ions, e.g. silver, gold, zinc, copper, magnesium) antimicrobial agents into the biomaterial surface. The antimicrobial agents can covalently be bounded to functionalized coatings, or incorporated into self-assembled mono/multilayer coatings during or after biomaterial production.
Antibiotics (e.g. vancomycin, daptomycin, rifampicin, amoxicillin, levofloxacin, gentamicin, or linezolid) are widely employed for the prevention and treatment of peri-prosthetic infections. Up to now, antibiotic-loaded implant materials are the only approach that has reached the market. However, this approach has some drawbacks namely: dose-dependent activity; sub-therapeutic delivery, limited diffusion into peri-implant tissues, and systemic and local cytotoxicity.
Antimicrobial peptides (AMPs) are an interesting group of anti-infective agents currently viewed as alternatives to mitigate the problem of antibiotic-resistant microorganisms. However, AMPs are expensive to produce and are vulnerable to pH fluctuations and proteases in the environment.
QS inhibitors show interesting antimicrobial properties, namely in inhibition and disintegration of biofilms. Nevertheless, some drawbacks are reported such as a narrow target spectrum, specifics QS for target bacteria, the lack of large-scale clinical QS inhibitors testing, and QS-resistant bacteria development.
Phage therapy has emerged as a potential alternative therapy to conventional antibiotics to manage and treat biofilm-related infections. However, more tests such as standardized and robust phage production, in vivo pharmacokinetic and pharmacodynamics studies, are required in order to allow phage therapy to become a standardized and well-accepted strategy in clinical practice.
Silver, gold, zinc oxide, titanium dioxide, magnesium oxide, or copper oxide are in-organic metallic nanoparticles (NPs), that have been deemed to be effective antimicrobial agents to fight against biofilm-related infections, however, there are still several side effects to be controlled concerning its use in clinical applications.
In sum, a variety of concepts and approaches are applied to create anti-infective bioengineered materials to combat infections associated with orthopedic biomaterials. However, there is still a long way to go before they are used in clinical settings, requiring it necessary to search for new approaches or improve the available ones in order to overcome safety and efficacy issues related to their implementation.”
Recent references were included into manuscript: 26, 27 and 28.

Reviewer 2 Report
The authors provided a review on the strategies to prevent and treat orthopedic biomaterial-related infections. The manuscript is well organized. However, to make it better, further improvements should be required by the following minor revision.
1. The authors need to add their own opinions and perspectives that incorporate the authors' insights in section 5.
2. In Table 1, the addition of relevant references is recommended.
3. The contents of Figure 1 should be explained in detail.
4. The addition of relevant new figures for several descriptions of previous works in section 4 is recommended to improve readability.
Author Response
Reviewer 2
Comment: The authors provided a review on the strategies to prevent and treat orthopedic biomaterial-related infections. The manuscript is well organized. However, to make it better, further improvements should be required by the following minor revision.
- The authors need to add their own opinions and perspectives that incorporate the authors' insights in section 5.
- In Table 1, the addition of relevant references is recommended.
- The contents of Figure 1 should be explained in detail.
- The addition of relevant new figures for several descriptions of previous works in section 4 is recommended to improve readability.
Answers for reviewer 2:
Dear reviewer
Thank you very much for your helpful and important comments. They were taken into consideration and responded to below and marked on the manuscript in yellow. Besides, some language improvements were made to the manuscript and were highlighted in yellow.
1 – The section 5 was modified according to your comment (Pages 13-14, section 5, lines 568-630, marked in yellow).
“Any surgical intervention, particularly ones including the implantation of biomaterials, has a high risk of associated infections. These infections can be devastating for the patients and overload healthcare systems.
Currently, prophylactic systemic antibiotic therapy is administered as a preventive and therapeutic measure on patients to whom an implant is applied. However, this therapy entails many disadvantages including the low drug concentration reaching the target site and the corresponding limited antimicrobial activity at the target site. Moreover, the wide variety of pathogenic bacteria causing infections related to orthopedic implants, the emergence of bacteria resistant to antibiotics and the cytotoxicity associated with them lead to the need to search for new alternative approaches to prevent and control implant-related infections.
The use of local strategies such as anti-adhesive surfaces or active surfaces has emerged as an effective approach to preventing and treating orthopedic biomaterial-related infections. Various concepts and approaches have been applied in the development of bioengineered materials with anti-infective properties, preventing microbial adhesion and colonization into bone tissue, and implant surfaces, and/or creating a free-bacteria environment around the implant. Biomaterials endowed with anti-infective properties need to be tailored according to the specific application and can be classified into two main groups: passive surfaces presenting chemistry and/or structure modifications which prevent or reduce microbial adhesion; and active surfaces with pre-entrapped or coated antimicrobial agents (antibiotics, peptides, bacteriophages, metals or metal ions), which will be released upon interaction with its surrounding environment and/or stimuli, killing the planktonic and sessile microorganisms.
A variety of polymers (e.g. PEG or polymer-based hydrogels), solutions (serum, plasma, or protein) and physical modifications (structure or morphology) can be used to create passive surfaces. Their anti-fouling mechanism is based on physicochemical mechanisms (e.g. steric repulsion, electrostatic repulsion, low surface energy, superhydrophobic and hydrophobic interactions, and substrate-microorganism physical interaction), which act as a physical barrier to avoid protein and bacterial adhesion.
Active biomaterials are based on the incorporation of organic (e.g. antibiotics, antimicrobial peptides, QS inhibitors, or bacteriophages) and inorganic (mainly metal ions, e.g. silver, gold, zinc, copper, magnesium) antimicrobial agents into the biomaterial surface. The antimicrobial agents can covalently be bounded to functionalized coatings, or incorporated into self-assembled mono/multilayer coatings during or after biomaterial production.
Antibiotics (e.g. vancomycin, daptomycin, rifampicin, amoxicillin, levofloxacin, gentamicin, or linezolid) are widely employed for the prevention and treatment of peri-prosthetic infections. Up to now, antibiotic-loaded implant materials are the only approach that has reached the market. However, this approach has some drawbacks namely: dose-dependent activity; sub-therapeutic delivery, limited diffusion into peri-implant tissues, and systemic and local cytotoxicity.
Antimicrobial peptides (AMPs) are an interesting group of anti-infective agents currently viewed as alternatives to mitigate the problem of antibiotic-resistant microorganisms. However, AMPs are expensive to produce and are vulnerable to pH fluctuations and proteases in the environment.
QS inhibitors show interesting antimicrobial properties, namely in inhibition and disintegration of biofilms. Nevertheless, some drawbacks are reported such as a narrow target spectrum, specifics QS for target bacteria, the lack of large-scale clinical QS inhibitors testing, and QS-resistant bacteria development.
Phage therapy has emerged as a potential alternative therapy to conventional antibiotics to manage and treat biofilm-related infections. However, more tests such as standardized and robust phage production, in vivo pharmacokinetic and pharmacodynamics studies, are required in order to allow phage therapy to become a standardized and well-accepted strategy in clinical practice.
Silver, gold, zinc oxide, titanium dioxide, magnesium oxide, or copper oxide are in-organic metallic nanoparticles (NPs), that have been deemed to be effective antimicrobial agents to fight against biofilm-related infections, however, there are still several side effects to be controlled concerning its use in clinical applications.
In sum, a variety of concepts and approaches are applied to create anti-infective bioengineered materials to combat infections associated with orthopedic biomaterials. However, there is still a long way to go before they are used in clinical settings, requiring it necessary to search for new approaches or improve the available ones in order to overcome safety and efficacy issues related to their implementation.”
2 – Relevant references were added according to your comment (Pages 3, Table 1, marked in yellow).
3 – In response to your comment, the content of Figure 1 was described in more depth (Page 4, lines 166-172, marked in yellow)
“The two initial phases involve the attachment of microorganisms via hydrophobic or electrostatic interactions to implant surfaces and cell-to-cell bindings. The microorganism growth and accumulation during the proliferation and maturation phases result in the development of a mature biofilm structure. Adhesive and disruptive processes occur during the biofilm maturation phase. The final stage of biofilm formation is the detachment phase, which involves microbial dispersal and dissemination, which may lead to new infection foci.”
4 - Unfortunately, it was not possible to obtain the authors’ authorization for Figures despite the requests made. Moreover, materials figure as raw materials won’t add any relevant information once most of the materials look very similar visually. These were the reasons that led us to choose to use schematics instead of images.

Reviewer 3 Report
This manuscript provides a thorough review on orthopedic biomaterials related-infections and on the possible solutions to prevent them. The topic is really up-to-date and addresses a key issue regarding the post-operative infections. The paper is relatively well organized and clear. The figures and tables are also informative and understandable. It summarizes well the current state on this research field and gives valuable contribution to the scientific literature.
Otherwise, my suggestion would be to supplement the manuscript with the Authors opinion on future prospectives and applicabilities of mentioned anti-infective, antifouling coatings/surfaces.
In general, the manuscript is appropriate for publication.
Author Response
Reviewer 3
Comment: This manuscript provides a thorough review on orthopedic biomaterials related-infections and on the possible solutions to prevent them. The topic is really up-to-date and addresses a key issue regarding the post-operative infections. The paper is relatively well organized and clear. The figures and tables are also informative and understandable. It summarizes well the current state on this research field and gives valuable contribution to the scientific literature.
Otherwise, my suggestion would be to supplement the manuscript with the Authors opinion on future prospective and applicability of mentioned anti-infective, antifouling coatings/surfaces.
In general, the manuscript is appropriate for publication.
Answers for reviewer 3:
Dear reviewer
Thank you very much for your helpful and important comments. They were taken into consideration and responded to below and marked on the manuscript in yellow. Besides, some language improvements were made to the manuscript and were highlighted in yellow.
1 – Additional information on the potential use and applicability of anti-infective, antifouling coatings and surfaces was added to section 5. (Pages 13-14, section 5, lines 568-630, marked in yellow).
“Any surgical intervention, particularly ones including the implantation of biomaterials, has a high risk of associated infections. These infections can be devastating for the patients and overload healthcare systems.
Currently, prophylactic systemic antibiotic therapy is administered as a preventive and therapeutic measure on patients to whom an implant is applied. However, this therapy entails many disadvantages including the low drug concentration reaching the target site and the corresponding limited antimicrobial activity at the target site. Moreover, the wide variety of pathogenic bacteria causing infections related to orthopedic implants, the emergence of bacteria resistant to antibiotics and the cytotoxicity associated with them lead to the need to search for new alternative approaches to prevent and control implant-related infections.
The use of local strategies such as anti-adhesive surfaces or active surfaces has emerged as an effective approach to preventing and treating orthopedic biomaterial-related infections. Various concepts and approaches have been applied in the development of bioengineered materials with anti-infective properties, preventing microbial adhesion and colonization into bone tissue, and implant surfaces, and/or creating a free-bacteria environment around the implant. Biomaterials endowed with anti-infective properties need to be tailored according to the specific application and can be classified into two main groups: passive surfaces presenting chemistry and/or structure modifications which prevent or reduce microbial adhesion; and active surfaces with pre-entrapped or coated antimicrobial agents (antibiotics, peptides, bacteriophages, metals or metal ions), which will be released upon interaction with its surrounding environment and/or stimuli, killing the planktonic and sessile microorganisms.
A variety of polymers (e.g. PEG or polymer-based hydrogels), solutions (serum, plasma, or protein) and physical modifications (structure or morphology) can be used to create passive surfaces. Their anti-fouling mechanism is based on physicochemical mechanisms (e.g. steric repulsion, electrostatic repulsion, low surface energy, superhydrophobic and hydrophobic interactions, and substrate-microorganism physical interaction), which act as a physical barrier to avoid protein and bacterial adhesion.
Active biomaterials are based on the incorporation of organic (e.g. antibiotics, antimicrobial peptides, QS inhibitors, or bacteriophages) and inorganic (mainly metal ions, e.g. silver, gold, zinc, copper, magnesium) antimicrobial agents into the biomaterial surface. The antimicrobial agents can covalently be bounded to functionalized coatings, or incorporated into self-assembled mono/multilayer coatings during or after biomaterial production.
Antibiotics (e.g. vancomycin, daptomycin, rifampicin, amoxicillin, levofloxacin, gentamicin, or linezolid) are widely employed for the prevention and treatment of peri-prosthetic infections. Up to now, antibiotic-loaded implant materials are the only approach that has reached the market. However, this approach has some drawbacks namely: dose-dependent activity; sub-therapeutic delivery, limited diffusion into peri-implant tissues, and systemic and local cytotoxicity.
Antimicrobial peptides (AMPs) are an interesting group of anti-infective agents currently viewed as alternatives to mitigate the problem of antibiotic-resistant microorganisms. However, AMPs are expensive to produce and are vulnerable to pH fluctuations and proteases in the environment.
QS inhibitors show interesting antimicrobial properties, namely in inhibition and disintegration of biofilms. Nevertheless, some drawbacks are reported such as a narrow target spectrum, specifics QS for target bacteria, the lack of large-scale clinical QS inhibitors testing, and QS-resistant bacteria development.
Phage therapy has emerged as a potential alternative therapy to conventional antibiotics to manage and treat biofilm-related infections. However, more tests such as standardized and robust phage production, in vivo pharmacokinetic and pharmacodynamics studies, are required in order to allow phage therapy to become a standardized and well-accepted strategy in clinical practice.
Silver, gold, zinc oxide, titanium dioxide, magnesium oxide, or copper oxide are in-organic metallic nanoparticles (NPs), that have been deemed to be effective antimicrobial agents to fight against biofilm-related infections, however, there are still several side effects to be controlled concerning its use in clinical applications.
In sum, a variety of concepts and approaches are applied to create anti-infective bioengineered materials to combat infections associated with orthopedic biomaterials. However, there is still a long way to go before they are used in clinical settings, requiring it necessary to search for new approaches or improve the available ones in order to overcome safety and efficacy issues related to their implementation.”
